# Catalytic amidation of natural and synthetic polyol esters with sulfonamides

Hua Liu[1], Yi-Ling Zhu[1] & Zhi Li [1]

Triacylglycerides are naturally abundant and renewable feedstock for biofuels and chemicals. In this report, these seemingly stable compounds are shown to be reactive toward a variety of sulfonamides under Lewis acid catalysis. In these reactions, alkyl $C(sp^3)$–O bonds are cleaved and C–N bonds constructed, providing functionalized value-added products directly from renewables. Mechanistic and scope study demonstrate that the origin of the reactivity could be the synergy of Lewis acid catalysis and neighboring group participation by the 2- or 3-acyloxy or acylamido group with respect to the reactive site. Since poly(ethylene terephthalate) (PET), a widely available consumer polyester, also contains 1,2-diol diester group as the repeating unit in the main chain, this chemistry can also be applied to efficient depolymerization of PET.

[1] School of Physical Science and Technology, ShanghaiTech University, Shanghai 201210, China. Correspondence and requests for materials should be addressed to Z.L. (email: lizhi@shanghaitech.edu.cn)

Biodiesel is an appealing source of renewable energy that received elevated attention in the recent decades[1]. Biojetfuel blends derived from plant oils reportedly reduced the emission of pollution particles comparing with traditional jet fuels[2]. Recently, gene engineering of oil-rich plants and microbes has been studied in order to boost production of plant oils[3–5]. Production of biodiesel commonly involves alkaline catalyzed transesterification of plant, animal, and microalgae oils, i.e. natural triacylglycerides (TAGs), with methanol to fatty acid methyl ester (FAMEs), a major component of biodiesel[6]. The transesterification process also produces an enormous amount of waste glycerol (projected to reach about 4 million tons per year by 2024)[7], often contaminated with water, alcohol, and alkaline, as an inevitable by-product. Containing a three-carbon chain, glycerol has been considered a very versatile platform for industrially meaningful chemicals[8,9]. Successful technology of glycerol valorization could revolutionize the chemical industry, shifting the production of many valuable industrial chemicals, such as 1,3-propanediol, allyl alcohol, acrylic acid, and many others, from relying on fossils to renewables[7,10–12]. But waste glycerol is hardly an ideal feedstock before costly purification[13]. The upcoming booming production of TAGs as projected above will make the situation even worse.

Since glycerol comes from TAGs, an intriguing question would be that how can TAGs be directly used as 3-carbon platform chemicals? Can we recover biodiesel + value-added products incorporated the 3-carbon motif while avoid generating waste glycerol? One such effort destroys the glyceryl motif before release: direct decarboxylative cracking and hydrotreatment of TAGs to hydrocarbons using heterogeneous catalysts affords products that can be used directly as diesel fuels[14–19]. These processes were typically performed at high temperatures (250–500 °C) and the C3 glyceryl motif was completely converted into propane which was too diluted to receive reasonable attention. A second reasonable transformation would be direct substitution of the acyloxy groups (RCOO–) of TAGs with other nucleophiles (NuH), constructing C–Nu functionality while recovering RCOOH. Surprisingly, although strategies that make use of synthetic glycerol derivatives were reported[20–22], few reports use TAGs directly as C3 building blocks for chemical production[23]. The reason might be that ester is usually considered as a rather stable functional group, especially the alkoxy $C(sp^3)$–O single bond[24]. In fact, in order to recover carboxylic acid that protected as an ester through nucleophilic cleavage of the C–O bond, some stringent conditions have to be met: high temperature (>100 °C), polar aprotic solvents (DMF, DMSO, and HMPA), stoichiometric salt of strong nucleophiles (halides, RSe–, RS–, RO–, and CN–), and C–O bonds that are labile towards $S_N1$ or $S_N2$ reactions (methyl, allyl, and benzyl. Fig. 1a)[25–27]. These procedures can surely be applied to a general scope of esters, but for glycol diesters or TAGs it would be difficult to selectively convert one acyloxy group while keeping the other ones unchanged.

Early works have shown alkyl esters, including TAGs, could undergo C–O hydrogenolysis to C–H and corresponding carboxylic acids catalyzed by tandem metal triflate Lewis acids and Pd hydrogenation catalyst (Fig. 1c)[28–32]. The C3 motif of glyceryl could be retained, but C–O hydrogenolysis of TAGs only showed moderate yield and selectivity toward highly valuable chemicals such as 1,2- and 1,3-propanediol derivatives[29]. Although acid-catalyzed C–O cleavage of mono-acyloxy group likely proceeds through an elimination-hydrogenation mechanism[31], the mechanism of TAGs hydrogenolysis remains elusive[29]. Diesters of 1,2- and 1,3-diols are known to undergo neighboring-group-assisted substitution of the acyloxy group through a cationic acyloxonium intermediate under stoichiometric Lewis acid-

mediated conditions, and so do TAGs[33]. Therefore, Lewis acid-catalyzed formation of acyloxonium intermediates and their catalytic hydrogenation could also be one of the operating mechanisms in the hydrogenolysis of TAGs.

In fact, acyloxonium cations (also named dioxolonium) are well-documented and can be generated from a number of ways, for examples (Fig. 1b): (1) alkene and iodine + silver ester salt (Prévost reaction[34] and Prévost-Woodward reaction[35]) or haloalkyl ester + silver salt[33]; (2) benzylidene acetal + NBS[36]; (3) protonation of ketene acetals[37]; and (4) diester + acid[33]. Breaking down acyloxonium intermediates with nucleophiles lead to many useful products described in the references above. Among the above strategies, silver salt strategies require stoichiometric silver salt and generate stoichiometric silver halides as by-products, while benzylidene/ketene acetals strategies require steps to synthesize these acetal precursors. On the other hand, the diester + acid method is a very promising strategy to perform acyloxonium chemistry in the regard of green chemistry, overcoming the abovementioned drawbacks of other counterpart strategies. First, many precursors containing a diester motif are naturally occurring (such as TAGs) or easily obtained from renewable resources (such as from sugar polyols and waste plastics). Second, carboxylic acid can be recovered as a by-product in this strategy. The biggest challenge here is catalysis in that among all the above methods, there has not been a catalytic strategy to generate and convert acyloxonium cations.

As implied in the TAGs hydrogenolysis work, strong triflate Lewis acids may also be able to catalyze the formation of acyloxonium cations, which would then participate in the selective ring-opening reactions, accomplishing catalytic substitution reactions of the acyloxy groups of TAGs, and even of many other 1,2- and 1,3-diol diesters. The investigation of the substitution reaction may lead to mechanistic understanding as well as alternative transformations of renewable feedstocks[38]. Herein, we present a remarkably simple catalytic reaction which directly substitutes the acyloxy groups of various polyol esters with arylsulfonamides, likely undergoing the acyloxonium intermediates (Fig. 1d). Such a transformation accomplishes the conversion of an ester $C(sp^3)$–O bond into more valuable C–N sulfonamide functionalities in not only TAGs, but also many other 1,2- and 1,3-diol diesters, even including esters of many natural polyols such as sugar alcohols and synthetic polyester polymers.

## Results

The substitutive amidation reaction of ethylene glycol diacetate (**1a**) with N-methyl-p-toluenesulfonamide (TsNHMe, **2a**) was chosen as a model reaction to investigate the reaction conditions. In the presence of different metal trifluoromethanesulfonate (triflate, OTf) catalysts $M(OTf)_n$ (2 mol %) in dichloroethane (DCE) at 120 °C for 24 h, mono-$C(sp^3)$-amidation product **3aa** was obtained as the only product. The yield largely depended on the Lewis acidity of metal triflates, among which $Hf(OTf)_4$ seemed to be the most efficient catalyst[39]. DCE was chosen as the solvent for initial screening because of its good solubility of reagents and catalysts. Note that the glycol diacetate was added in an excess (2 equiv.) to facilitate better conversion of the sulfonamide nucleophile. The reaction also went much faster under the solventless condition, giving 90% yield of **3aa** after heating for just 6 h and 95% for 24 h (Supplementary Table 1). The results of reactions were similar when different acyl groups were investigated (Table 1, entries 1–4). But triacetin and tributyrin (Table 1, entries 5–6) could not accomplish full conversion (~60% conversion of **2a**) even with the increase of catalyst loading (5 mol%), affording only the mono-amidation products in moderate yields.

**Fig. 1** Substitution at C(sp³)–O bonds of esters. **a** Classical S$_N$2 substitution reactions at the alkyl C–O bond of an ester; **b** Examples of reactions involving acyloxonium intermediates; **c** Catalytic hydrogenolysis of triacylglycerides; **d** Catalytic substitution of polyol esters with sulfonamide. R, R¹, R², R′, dash bonds: substituents; Nu, nucleophile; M, metal

Note that reactions of triglycerides selectively afforded mono-amidation products with only very small amount of 1,3-diami-dation products (<5%), while no substitution took place at the secondary acyloxy group.

Different sulfonamide nucleophiles were next evaluated. TsNH₂ (**2b**) reacts with **1a** smoothly to give 85% of the mono-amidation product in just 14 h, which further reacts with **1a** once more to afford *N,N*-dialkylated sulfonamide in 10% yield (Table 1, entry 7). Amides and carbamides were not reactive at all (Table 1, entries 8–10). *N*-methyl methanesulfonamide and *N*-methyl benzenesulfonamides bearing electron-withdrawing groups or electron-donating groups at the benzene ring all proved to be good nucleophiles, providing the corresponding mono-amidation product as the single product in excellent yields (Table 1,

**Table 1 Reaction Conditions Optimization for Polyol Esters and Nucleophiles[a]**

| | Polyol ester, 1 | Nucleophile, 2 | Catalyst | mol % | Temp /°C | Yield /% |
|---|---|---|---|---|---|---|
| 1 | Glycol diacetate (**1a**) | TsNHMe (**2a**) | Hf(OTf)$_4$ | 2 | 120 | 90 |
| 2 | Glycol dipropionoate (**1b**) | TsNHMe (**2a**) | Hf(OTf)$_4$ | 2 | 120 | 83 |
| 3 | Glycol diisobutyrate (**1c**) | TsNHMe (**2a**) | Hf(OTf)$_4$ | 2 | 120 | 84 |
| 4 | Glycol dipivalate (**1d**) | TsNHMe (**2a**) | Hf(OTf)$_4$ | 2 | 120 | 83 |
| 5 | Triacetin (**1e**) | TsNHMe (**2a**) | Hf(OTf)$_4$ | 5 | 120 | 42 |
| 6 | Tributyrin (**1f**) | TsNHMe (**2a**) | Hf(OTf)$_4$ | 5 | 120 | 48 |
| 7[b] | Glycol diacetate (**1a**) | TsNH$_2$ (**2b**) | Hf(OTf)$_4$ | 2 | 120 | 85 |
| 8 | Glycol diacetate (**1a**) | AcNHMe (**2c**) | Hf(OTf)$_4$ | 2 | 120 | 0 |
| 9 | Glycol diacetate (**1a**) | BocNHMe (**2d**) | Hf(OTf)$_4$ | 2 | 120 | 0 |
| 10 | Glycol diacetate (**1a**) | Phthalimide (**2e**) | Hf(OTf)$_4$ | 2 | 120 | 0 |
| 11 | Glycol diacetate (**1a**) | CH$_3$SO$_2$NHMe (**2f**) | Hf(OTf)$_4$ | 2 | 120 | 92 |
| 12 | Glycol diacetate (**1a**) | PhSO$_2$NHMe (**2g**) | Hf(OTf)$_4$ | 2 | 120 | 94 |
| 13 | Glycol diacetate (**1a**) | 4-BrPhSO$_2$NHMe (**2h**) | Hf(OTf)$_4$ | 2 | 120 | 95 |
| 14 | Glycol diacetate (**1a**) | 4-CF$_3$PhSO$_2$NHMe (**2i**) | Hf(OTf)$_4$ | 2 | 120 | 98 |
| 15 | Glycol diacetate (**1a**) | 4-MeOPhSO$_2$NHMe (**2j**) | Hf(OTf)$_4$ | 2 | 120 | 91 |
| 16 | Glycol diacetate (**1a**) | 4-NO$_2$PhSO$_2$NHMe (**2k**) | Hf(OTf)$_4$ | 2 | 120 | 0 |
| 17 | Glycol diacetate (**1a**) | 4-AcNHPhSO$_2$NHMe (**2l**) | Hf(OTf)$_4$ | 2 | 120 | 0 |
| 18 | Glycol diacetate (**1a**) | Saccharin (**2m**) | Hf(OTf)$_4$ | 2 | 120 | 98 |
| 19[c] | Tributyrin (**1f**) | Saccharin (**2m**) | Hf(OTf)$_4$ | 2 | 120 | 60 |
| 20[c] | Tributyrin (**1f**) | Saccharin (**2m**) | Hf(OTf)$_4$ | 5 | 120 | 66 |
| 21[c] | Tributyrin (**1f**) | Saccharin (**2m**) | Yb(OTf)$_3$ | 5 | 120 | 40 |
| 22[c] | Tributyrin (**1f**) | Saccharin (**2m**) | Al(OTf)$_3$ | 5 | 120 | 45 |
| 23[c] | Tributyrin (**1f**) | Saccharin (**2m**) | Sc(OTf)$_3$ | 5 | 120 | 76 |
| 24[c] | Tributyrin (**1f**) | Saccharin (**2m**) | Sc(OTf)$_3$ | 5 | 150 | 85 |
| 25[c,d] | Tributyrin (**1f**) | Saccharin (**2m**) | Sc(OTf)$_3$ | 2 | 150 | 90 |
| 26[c,e] | Tributyrin (**1f**) | Saccharin (**2m**) | Sc(OTf)$_3$ | 2 | 150 | 66 |
| 27[c,f] | Tributyrin (**1f**) | Saccharin (**2m**) | Sc(OTf)$_3$ | 2 | 150 | 65 |

*Tf* trifluoromethanesulfonyl, *Ts* 4-methylbenzenesulfonyl, *Ac* acetyl, *Boc* t-butyloxycarbonyl, *neat* without solvent
[a]**1** (1.0 mmol), **2** (0.5 mmol) and catalyst were stirred at 120 °C for 24 h; Yields are isolated unless otherwise noted
[b]Reaction time: 14 h. *N,N*-dialkylated product 10% obtained
[c]NMR yield
[d]85% isolated yield, along with 3% double substitution
[e]1:1 of tributyrin and saccharin, product contains 14% double substitution product
[f]1:2 of tributyrin and saccharin, product contains 25% double substitution product

entries 11–15). But *N*-methyl-4-nitrobenzenesulfonamide and *N*-methyl-4-acetamidobenzenesulfonamide gave no product at all (Table 1, entry 16–17). Surprisingly, saccharin (**2m**), a sulfonamide that widely used as an artificial sweetener in food industry as well as a nitrogen source in catalytic diamination reaction of alkenes[40,41], is shown to be a superior nucleophile than others. When examined in the glycol diacetate amidation reaction, almost quantitative yield of **3am** was obtained, possibly because the higher acidity of saccharin than TsNHMe allows higher concentration of sulfonamidate anion, which could be the real nucleophile (Table 1, entry 18).

Intrigued by the improved performance of saccharin, we then investigated the reactivity between saccharin and tributyrin, a representative TAG. Mono-substituted product was obtained in 60% yield under the same conditions as above, while increasing the loading of Hf(OTf)$_4$ to 5 mol % did not significantly improve the yield (Table 1, entries 19–20). Among several metal triflates, Sc(OTf)$_3$ gave significantly higher yield, 76%, which could be further improved to 85% by raising the reaction temperature to 150 °C (Table 1, entries 21–24). Interestingly, lowering the loading of Sc(OTf)$_3$ to 2 mol % could afford a higher yield of the product, probably by

reducing the double amidation side-reaction (Table 1, entry 25). Indeed, when the stoichiometry of saccharin was increased, higher yields of the double substitution product were obtained (Table 1, entries 26–27).

These results inspired us to further explore the scope of polyol esters (Fig. 2). A series of TAGs derived from different aliphatic and aromatic carboxylic acids were converted to the corresponding mono-sulfonamidation products (**3em–3jm**) in good yields, along with recovered carboxylic acids. In particular, tricaprylin (**1i**) required stronger conditions (5 mol % catalyst, 180 °C, 24 h) than the standard condition, giving only 72% yield. On the other hand, tristearin (**1j**), major component of animal fats, gave mono-amidation product **3jm** in 79% yield after heating for 48 hours. In a gram-scale example product **3em** was obtained in 80% yield. Subjecting the resulting mono-amidation products in the reaction conditions again gave double-amidation products **3emm** and **3fmm** in good yields. Then a series of 1,2-diacetyloxy alkanes were examined. They all gave good yields of the mono-amidation products (**3km–3om**), although longer alkyl side chains required longer reaction time and higher stoichiometry of the ester to achieve high yields, likely due to the competitive elimination of the secondary acetyloxy group along with its β-H.

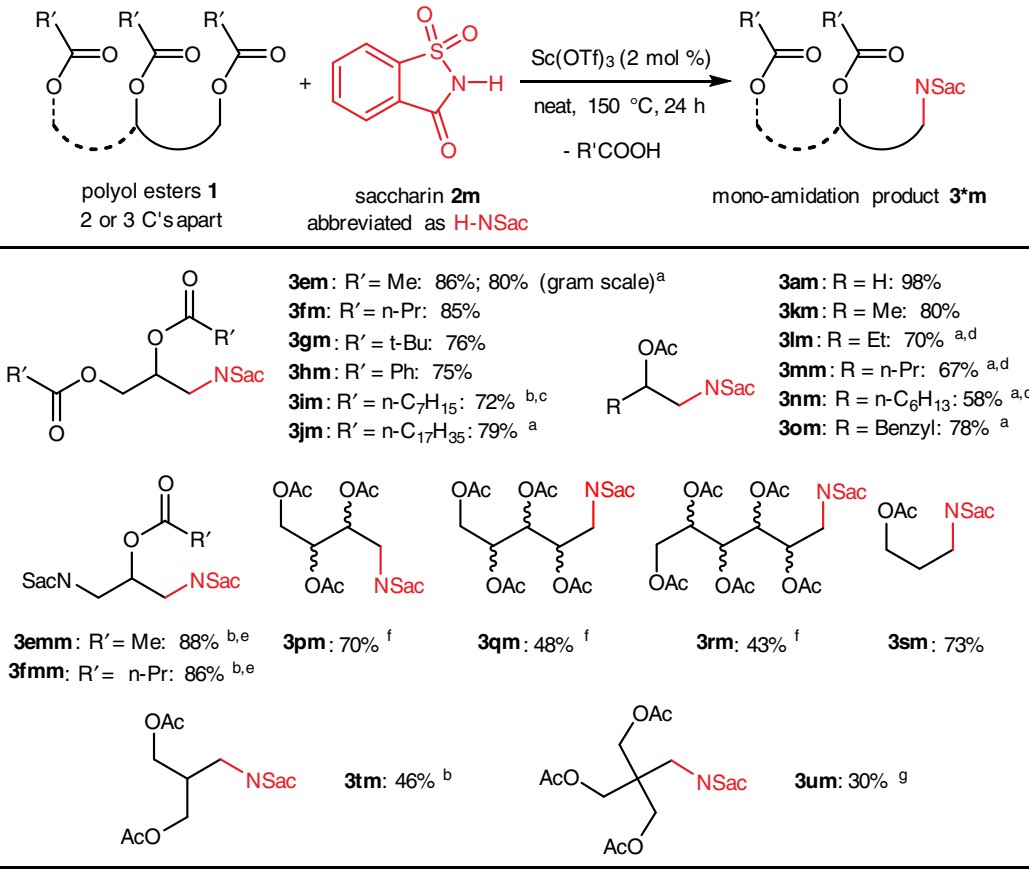

**Fig. 2** Scope of polyol esters mono-amidation with saccharin. Reaction conditions: **1** (1.0 mmol), **2m** (0.5 mmol) and Sc(OTf)$_3$ (2 mol %) was stirred at 150 °C for 24 h unless otherwise noted; isolated yields; [a]48 h; [b]5 mol % of Sc(OTf)$_3$; [c]180 °C; [d]2.0 mmol of **1** was used; [e]starting material is mono-amidation product; [f]mixture of stereoisomers; [g]5 mol % of Hf(OTf)$_4$ was used, and a drying tube filled with K$_2$CO$_3$ and Na$_2$SO$_4$ was equipped to assimilate AcOH during reaction. Wavy bonds = undefined stereochemical configuration

We then took the challenge of molecules containing even more ester groups than TAGs. For example, erythritol, xylitol, and sorbitol are all well-known sugar-derived polyols, as well as tough targets for biomass conversion[42–44]. Sugar-derived aminopolyols are also potentially useful molecules in drug discovery because of their structural resemblance to carbohydrates, and they are extensively studied synthetic targets[45–50]. In our strategy, all hydroxyl groups of erythritol, xylitol, and sorbitol were first fully acetylated, and the resulting esters (**1p**, **1q**, and **1r**) were subjected to the standard catalytic sulfonamidation conditions with saccharin. Terminally mono-amidation products (**3pm**, **3qm**, and **3rm**) were obtained as mixtures of stereoisomers in moderate yields. Note that the starting materials were all single stereoisomers, but the stereocenters of the products were all scrambled (See Supplementary Fig. 1 for erythritol acetate epimerization).

Moreover, 1,3-diesters were also examined. 1,3-Propanediol diacetate (**1s**) gave mono-amidation product in 73% yield, demonstrating that acyloxy groups in 1,3- relationship were also reactive. Tris(hydroxymethyl)methane triacylate (**1t**) and pentaerythritol tetraacylate (**1u**) are interesting substrates because they both contain multiple 1,3-relationships hence might allow multiple substitution like TAGs. However, they both gave exclusively mono-amidation products (**3tm** and **3um**) in low yields and incomplete conversion of reagents, even after repeated reactions.

The exclusive generation of mono-amidation products strongly suggested that the neighboring group participation mechanism is probably operating in these reactions (Fig. 3a). In contrast to the facile reactions of 1,2- and 1,3-diesters, 1,4-butanediol diacetate (**1v**) gave <5% yield of mono-amidation, while phenylpropyl acetate (**1w**), an isolated ester, gave 18% yield (Fig. 3b). These observations consolidated the neighboring group participation effect which was more pronounced when a five- or six-membered cyclic intermediate could be generated by an intramolecular substitution[51,52]. We then synthesized and isolated the [BF$_4$]$^-$ salt of the cationic intermediate in stoichiometric amount[33]. Reacting this salt with saccharin afforded the mono-amidation product **3am** in good yield, demonstrating that the mechanism likely undergoes this proposed pathway (Fig. 3c). Lastly, the stereochemistry scrambling phenomena of sugar alcohol esters could also be explained by this mechanism (Fig. 3d). Generation of the cationic intermediate may come from either the terminal acyloxy group attacking an internal acyloxy group (red arrows) or the opposite way (blue arrows). Then before the nucleophile approaches, the leaving acyloxy group may rebound to either the primary or the secondary carbon of the five-membered ring intermediate, resulting stereochemistry inversion. This process seems to be a fast equilibrium under Lewis acid catalysis, thus it would possibly result in a thermodynamic mixture of stereoisomers at the end (See also the supporting information for an experiment on erythritol tetraacetate epimerization without nucleophile). These results implied that both the terminal/primary and internal/secondary alkoxy C–O bonds were being activated by the Lewis acid as a leaving group, but only the terminal/primary position of the

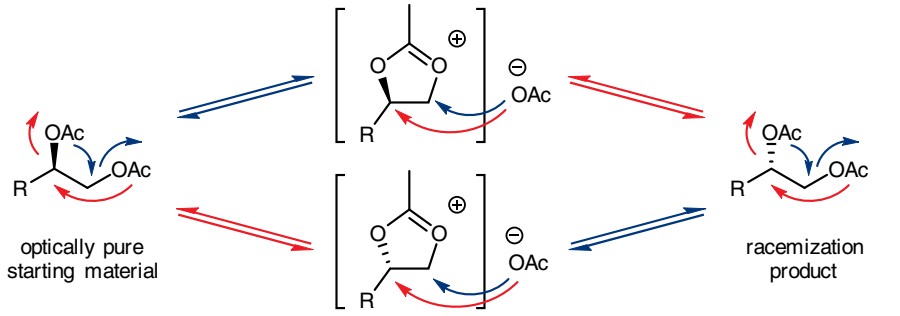

**Fig. 3** Proposed mechanism and experimental evidence. **a** Proposed mechanism of mono-amidation of diol diesters; **b** Results of neighboring group participation experiments; **c** Stoichiometric ring-opening of acyloxonium salt with saccharin; **d** pathways of stereocenters scrambling through acyloxonium intermediates. LA , Lewis acid; rt , room temperature

cyclic cationic intermediate was susceptible for nucleophilic attack by saccharin.

It is then conceivable that the neighboring participation group might not have to be an ester carbonyl. Instead, an amide carbonyl would be more nucleophilic while less labile as a leaving group. Hence, a molecule containing both amide and ester functional groups in a suitable distance should produce an amide sulfonamide, a diamine derivative, under this catalytic substitution. Investigations were performed on a series of amide esters, which were easily prepared from amino alcohols (Fig. 4). Brief conditions screening on 2-acetamidoethyl acetate (**4a**) revealed that Hf(OTf)₄ and 150 °C are effective conditions (Supplementary Table 2). Amide sulfonamide **5am** was obtained in an excellent yield of 92% under the optimized condition, as well as those from substrates bearing substituents at the α- and β-positions of the acyloxy group (**5bm ~ 5gm**). In these cases, the acyloxy group acts exclusively as the leaving group, while the amide acts exclusively as the intramolecular nucleophile. Thus, stereochemistry of the parent amino alcohols is retained (See Supplementary Fig. 2 for high-performance liquid chromatography

results, and Supplementary Fig. 3 and Supplementary Tables 3 for X-ray crystallography structure of (R)-**5dm**). When two equivalent acyloxy leaving groups were present, selectivity of monoamidation could be achieved by using an excess of the substrate (**5hm**). Compounds **5im** and **5jm** were observed as the rotational isomer mixtures about the amide bond. Amide at the γ- position of the acyloxy group is also an effective directing group (**5km**). The stereochemistry outcome of the reaction of exocyclic amide esters was quite intriguing: the same relative *trans-* configuration of **5lm** was always obtained, regardless of whether the reaction started from either the *cis-* or *trans*-1-acetamido-2-indanol acetate (**4l** and **4l'**). Similar results were also observed in the reaction from *trans*-2-acetamido-2-cyclopentanol acetate (**4m**) to the *trans-* amide sulfonamide **5mm**. There might be a common fused ring cationic transition state in these reactions that gives the *trans-* products while performing ring-opening.

Lastly, we expand the scope of this chemistry beyond natural polyol esters into synthetic polyesters. Polyesters, in particular, poly(ethylene terephthalate) (PET), are an essential class of materials for modern society. Since PET is widely used in

**Fig. 4** Scope of amide esters mono-amidation with saccharin. Reaction conditions: **4** (0.6 mmol), **2m** (0.5 mmol) and Hf(OTf)$_4$ (1.0 mol %) was stirred at 150 °C for 24 h unless otherwise noted; isolated yields; [a]1.0 mmol (2.0 equiv) of **4h** was used. Product mixture contains 18% double substitution product

consumer products such as containers and textiles, rapid accumulation of post-consumer PET wastes has been an emerging environmental issue. Efforts for PET recycling include physical methods such as remolding, and chemical depolymerization, such as hydrolysis[53], transesterification[54,55] hydrogenation[56], and catalytic hydrosilation[57]. These chemical methods all rely on carbonyl chemistry. On the other hand, PET units are ethylene glycol diesters, excellent potential precursors for the Lewis acid-catalyzed amidation strategy described above. Indeed, upon catalytic substitution by saccharin, PET pieces directly cut from an empty plastic soda bottle were completely degraded to *para*-terephthalic acid (PTA) and sulfonamide products (Fig. 5a). Note that toluene was used as solvent to facilitate the diffusion of the catalyst and saccharin into the polymer body. Compound **8** was precipitated from the addition of 0.5 M NaOH aqueous solution into the product mixture in 86% yield (Fig. 5b). Basic PTA and trace amount of mono-amidation products (such as **7**) are still in the aqueous phase. In order to recover PTA as much as possible, a one-pot substitution-hydrolysis protocol was conducted: after the catalytic substitution was done, 6 M NaOH aq was added to the reaction mixture and stirred at 60 °C for 3 h. Acidification by HCl rapidly precipitates PTA out in almost quantitative yield (97%), while the sulfonamido alcohol **9** was extracted out from the aqueous phase in 90% yield. The production of **9** is a decisive evidence that the PET chain is cleaved by the catalytic substitution chemistry rather than carbonyl chemistry. A two-gram scale demonstration was also performed, in which 95% PTA and 84% of **9** were recovered while consuming less solvent and base per unit mass of products obtained than that of the small-scale reaction. This one-pot strategy for PET depolymerization showed clear advantages over other protocols: (1) complete depolymerization under mild condition; (2) easy work-up and isolation of products; and (3) production of value-added chemicals containing sulfonamide functionality, other than the usual ethylene glycol product.

## Discussion

Some of the sulfonamide products obtained in this research are sporadically reported in the literature and patents as a variety of functional molecules or their synthetic precursors. For example, some N-hydroxyethyl saccharin derivatives were reported to exhibit biological and medical activities[58–60], some 1-aminoglycerol sulfonamides were reported to serve as components in printing ink[61]. Nevertheless, deprotection of sulfonamides is very challenging chemistry, which usually requires drastic conditions and stoichiometric hazardous reagents[62]. We tested reported procedures for saccharin deprotection[41] but none of them gave satisfactory efficiency obtaining the free amine product. In this regard, future direction of this research would focus on development of catalysts and more easily cleavable or useful nucleophiles rather than deprotection of sulfonamide groups. Meanwhile, applications of the current sulfonamide products are also being studied.

## Methods

**General catalytic procedures for polyol esters**. Polyol ester (1.0 mmol), sulfonamide (0.5 mmol) and Hf(OTf)$_4$ (0.01 mmol) were added to a 5 mL sample vial equipped with a magnetic stir bar. The vial was sealed and stirred at 120 °C for 24 h. The mixture was then cooled to room temperature and purified by column chromatography (PE: EA = 4:1).

**General catalytic procedures for amide esters**. Amide esters (0.6 mmol), saccharin (0.5 mmol) and Hf(OTf)$_4$ (0.005 mmol) were added to a 5 ml sample vial equipped with a magnetic stir bar. The vial was sealed and stirred at 150 °C for 24 h. The mixture was then cooled to room temperature and purified by column chromatography (PE: EA = 1:1).

**Gram scale one-pot depolymerization of PET**. PET pieces (1.92 g, 10 mmol), saccharin (1.83 g, 10 mmol) and Hf(OTf)$_4$ (350 mg, 0.5 mmol) were added to a 50 mL reaction tube equipped with a magnetic stir bar, followed by the addition of toluene (10 mL). The tube was then sealed and stirred at 150 °C for 48 h. After cooling to room temperature, a solution of NaOH (2.40 g, 60 mmol) and water (20 mL) was added to the reaction tube. The resulting mixture was further stirred at 60 °C for 12 h, and then partitioned between water (50 mL) and ethyl acetate (50 mL). The aqueous

**Fig. 5** Depolymerization of PET. **a** General one-pot reaction sequence of Lewis acid-catalyzed depolymerization of PET with saccharin. Recovered yields of terephthalic acid **6** and byproduct **9** were shown for reactions run at 1 mmol scale and 2 g (10 mmol) scale. **b** Analysis of solids precipitated out from the intermediate reaction mixture. aq, aqueous solution

phase was separated and acidified to pH ~ 4 by dropwise addition of 12 M HCl solution to precipitate out terephthalic acid. The solid was then filtered, washed with water (20 mL), and dried under vacuum. 1.57 g of terephthalic acid was obtained as a white solid in 95% yield. The resulting aqueous filtrate was extracted with ethyl acetate (100 mL*3). The combined organic phase was washed with saturated NaCl solution (100 mL), dried with $Na_2SO_4$ and evaporated under vacuum. 2.05 g of compound **9** was obtained as a white solid in 84% yield.

## Data availability

The authors declare that all data supporting the findings of this study are available within the article and its Supplementary Information files. Extra data are available from the corresponding author upon request. Supplementary Tables and Figures, experimental procedures, characterization and spectra of all materials, and X-ray crystallographic coordinates data of (*R*)-**5dm** are included in the Supporting Information. The X-ray crystallographic coordinates for (*R*)-**5dm** have also been deposited at the Cambridge Crystallographic Data Centre (CCDC), under deposition numbers CCDC 1892474. These data can be obtained free of charge from The Cambridge Crystallographic Data Centre via www.ccdc.cam.ac.uk/data_request/cif.

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

## Acknowledgements

Financial support for this work was generously provided by the National Natural Science Foundation of China (Grant No. 21673141), "1000 Talents Plan for Young Professionals" Start-up funding, and ShanghaiTech University start-up funding. We thank Shanghai Advanced Research Institute, CAS for supporting lab space, and Postdoctoral Office of the Shanghai Institute of Organic Chemistry, CAS for supporting H.L. We specially thank Prof. Tao Li and Dr. Na Yu for performing crystal XRD analysis, and Mr. Xiao-Long Xu for the help in measuring melting points. We also thank other staff members at Analytical Instrumentation Center of SPST, ShanghaiTech for characterization support: Dr. Huawei Liu and Dr. Min Peng for NMR, Dr. Min Yang for HRMS, and Dr. Lili Du for elemental analysis.

## Author contributions

Z.L. conceived the idea and supervised the work. H.L. and Z.L. designed the experiments. H.L. performed all materials preparations, catalytic reactions and depolymerization reactions. H.L. and Y.-L.Z. performed mechanistic experiments. All authors contributed to data analysis and manuscript writing.

## Additional information

**Competing interests:** Z.L. and H.L. are inventors of three pending patent applications. The remaining author declares no competing interests.

