## [Peer Review File · Nature Communications]

REVIEWERS' COMMENTS:

Reviewer #1 (Remarks to the Author):

Having read the original reviewer reports and the response from the authors and examined the manuscript, I feel that the authors have made considerable efforts to address the criticisms made and to provide additional data/discussion where requested. The paper reports a very interesting new approach to the functionalisation of renewable feedstocks and of waste plastic which should prove practically useful and could also stimulate further research in this area. The procedure is effective and works with a wide range of substrates. I am very happy to recommend publication of the manuscript in its current form though there is a need for improvement of the grammar/style in some sections during the editing process.

Response to Reviewer's comments **in blue**:

Reviewer #1 (Remarks to the Author):

Having read the original reviewer reports and the response from the authors and examined the manuscript, I feel that the authors have made considerable efforts to address the criticisms made and to provide additional data/discussion where requested. The paper reports a very interesting new approach to the functionalisation of renewable feedstocks and of waste plastic which should prove practically useful and could also stimulate further research in this area. The procedure is effective and works with a wide range of substrates. I am very happy to recommend publication of the manuscript in its current form though there is a need for improvement of the grammar/style in some sections during the editing process.

We highly appreciate the Reviewer 1's precise understanding of our chemistry, and supportive and favorable comments.